# The Maternal Diet with Fish Oil Might Decrease the Oxidative Stress and Inflammatory Response in Sows, but Increase the Susceptibility to Inflammatory Stimulation in their Offspring

**DOI:** 10.3390/ani10091455

**Published:** 2020-08-19

**Authors:** Wenli Luo, Weina Xu, Jing Zhang, Jianbo Yao, Jianxiong Xu

**Affiliations:** 1Shanghai Key Laboratory of Veterinary Biotechnology, School of Agriculture and Biology, Shanghai Jiao Tong University, Shanghai 200240, China; maple_rowe@sjtu.edu.cn (W.L.); xuweina@sjtu.edu.cn (W.X.); zhangjing224@sjtu.edu.cn (J.Z.); 2Division of Animal and Nutritional Sciences, West Virginia University, Morgantown, WV 26506, USA; Jianbo.Yao@mail.wvu.edu

**Keywords:** sow-reared piglet, ROS, LPS, cytokine, liver

## Abstract

**Simple Summary:**

Fish oil is rich in long-chain n-3 polyunsaturated fatty acids (n-3 LC-PUFA), which play an important role in the regulation of oxidative stress and inflammatory response. In the present study, increasing n-3 LC-PUFA in the maternal diet with fish oil from the 84th day of gestation until the 16th day of lactation decreased the oxidative stress and inflammatory response in sows and enhanced the parameters related to the antioxidative capacity. However, the inflammatory response of suckling piglet increased pre-/post-lipopolysaccharide (LPS) challenge. We concluded that the maternal diet with fish oil might decrease the oxidative stress and inflammatory response in sows, and enhance the antioxidative ability but increase the susceptibility to inflammatory stimulation in their progenies.

**Abstract:**

The aim of this study is to investigate the effect of the maternal diet with fish oil on the oxidative stress and inflammatory response in sows, and the protective effect on the piglets suckling the sows fed the diet with fish oil in the context of inflammatory stimulation. Twelve sows were divided into two groups. Sows were fed soybean oil diet (SD) or soybean oil + fish oil diet (FD) from gestation to lactation period. The blood samples of sows were collected from the auricular vein at the 16th day of lactation. One piglet was selected from each litter on the 14th day after birth. Lipopolysaccharide (LPS) was injected into the neck muscle after pre-treatment blood samples were collected from the anterior vena cava of piglets. The blood samples of piglets were collected at 5 h and 48 h post-LPS injection from the front cavity vein. Liver samples were collected at 48 h post-LPS injection. The FD diet significantly increased the level of high-density lipoprotein cholesterol (HDL-C) in the plasma of lactating sow, decreased the levels of alkaline phosphatase(AKP) and tumor necrosis factor alpha(TNF-α) in the plasma of lactating sows, and increased the level of immunoglobulin G(IgG) in the colostrum and interleukin-10(IL-10) in the milk (*p* < 0.05). In the FD group, the levels of glutathione peroxidase (GSH-Px) and total antioxidant capacity (T-AOC) significantly increased in the plasma of piglets at 48 h post-LPS injection (*p* < 0.05). Meanwhile, the relative expression of GSH-Px mRNA was decreased in the FD group (*p* < 0.05). However, the levels of interleukin-1 beta (IL-1β) and interleukin-6(IL-6) in the plasma of piglets were significantly higher in the FD group pre- and post-LPS injection (*p* < 0.05). The ratio of the phosphonated extracellular regulated protein kinases to the extracellular regulated protein kinases (p-ERK/ERK) protein in the livers of piglets was decreased (*p* < 0.05), but the expression of nuclear transcription factor-κB (NF-κB) mRNA and the ratio of the phosphonated inhibitor of NF-κB to the inhibitor of NF-κB (p-IκB-α/IκB-α) protein was increased in the livers of piglets (*p* < 0.05). These results indicate that a maternal diet with fish oil might decrease the oxidative stress and inflammatory response in sows, and enhance the antioxidative ability but increase the susceptibility to inflammatory stimulation in their progenies.

## 1. Introduction

Neonatal piglet survival is the major problem in the modern pig industry, because the imperfect immune system of neonatal piglet makes it sensitive to the pathogen. Neonate piglets must acquire maternal immunoglobulins from ingested colostrum for passive immune protection, before they adequately produce their own immunoglobulins at approximately 3–4 weeks of age [1]. The liver plays a central role in the regulation of lipid metabolism and contains numerous innate and adaptive immune cells that specialize in the detection and capture of pathogens from the blood [2]. Failure of the liver to either detect or clear pathogens can result in systemic infections, often leading to death, clearly highlighting the critical importance of the liver in the maintenance of host immunity [3]. Nutrition during the gestation to lactation period may have a fundamental impact on fetal immune development [4,5]. Fish oil is rich in long-chain n-3 polyunsaturated fatty acids (n-3 LC-PUFA), such as 20:5n-3 (EPA) and 22:6n-3 (DHA), which are essential fatty acid for pig and exert beneficial anti-inflammatory effects in animal and cell models [6]. The colostrum and milk from the sows are the main source of nutrient for the suckling piglets. The lactating diet of a sow with 3–5% fish oil is beneficial for the health of suckling piglet by changing the fatty acid composition, increasing the immunoglobulins and/or decreasing the level of cytokines in the milk [7,8,9]. Therefore, maternal nutrition might be associated with the immunomodulation of the progeny. So far, the effect of the maternal diet with fish oil on the immunomodulation of the progeny in the context of LPS-induced inflammation is unclear. In our previous study, the maternal diet with fish oil prolonged the gestational duration, alleviated oxidative stress in sows on the farrowing day and modulated inflammatory response in sows and their offspring [10]. In this study, we hypothesized that the maternal diet with fish oil could change the inflammatory response of the sows and potentiate the resistance to inflammation in the suckling piglets under the stimulation of immune stress.

The objective of this study was to investigate the effects of the maternal diet with fish oil on the oxidative stress and inflammatory response in sows, as well as on the oxidative stress and inflammatory response in their offspring, before and after LPS challenge.

## 2. Materials and Methods

All experimental protocols were approved by the Animal Care and Use Committee of the Shanghai Jiaotong University (No. SYXK 2013-0052). The study took place at the experimental study farm of a feeding company (Xinnong Feed Ltd., Shanghai, China) during May and June 2016.

### 2.1. Animal, Diet and Animal Management

Twelve second-parity sows (hybrid Topigs 20 breed sows, Dutch Landrace × Great York) and their piglets [(Dutch Landrace × Great York) × Duroc] were used in the study. On the 84th day of gestation, the back fat thickness of sows was measured, and 12 sows who had similar back fat thickness (soybean oil group: 15.14 ± 0.51 vs. fish oil group: 15.43 ± 0.48 mm; *p* = 0.69) were selected for our study. Sows were inseminated with semen from the same Duroc boars and the expected delivery day within one week. From the 84th day of gestation until the 16th day of lactation, 12 sows were divided into two dietary treatment groups: One group was fed the soybean oil maternal diet (SD), and the other group was fed the fish oil supplemented diet (FD) during the experimental period. The fatty acid composition of the soybean oil 100 (Four Grade GB 1535–2003, Yihai Kerry Group) and fish oil (Product name: refined fish oil, 101 Batch number: 811393, NovoSana Co. Ltd., Taicang, China) was analyzed before use (Appendix A). Diets were formulated according to the sow’s nutrient requirements from the National Research Council (NRC, 2012) [11]. Diet formulations are as reported in Luo et al. (2019) [12]. The fatty acid composition in experimental diets was determined as described by Raes et al. [13], and was reported in Luo et al. (2019) [12]. All diets were mash feed and were stored in vacuum dark storage bags (20 kg/bag), and kept in 24–28 °C constant temperature warehouse before use.

Within 24 h post-farrowing, litter size was equalized by cross-fostering to achieve 12–14 piglets per sow within the same treatment group. All suckling piglets were housed in corresponding farrowing unit with incubator and heat lamp for piglets from farrowing day to weaning day. Piglets were not fed the standard creep feed before day 16. Sows and piglets had free access to water.

### 2.2. LPS Treatment of Piglets

On the 14th day after birth, twelve piglets (three males and three females from each group) were selected in the study. One piglet per litter was selected, and the bodyweight of the selected piglets is close to the average bodyweight of the litter (Initial Weight: SD = 4.12 ± 0.05 kg vs FD = 4.11 ± 0.03 kg; *p* = 0.87). Blood samples (5 mL) were collected from each selected piglet, and then all selected piglets were administrated via the cervical side behind the left ear with E. coli LPS at 80 μg/kg BW. The LPS (Escherichia coli serotype 055: B5, Sigma Chemical, St. Louis, MO 63103, USA) was dissolved in sterile 0.9% NaCl solution (500 mg LPS per liter of saline). Blood samples (5 mL) were collected from each piglet at 5 h and 48 h post-LPS challenge. All piglets were weighed and recorded before slaughtering at 48 h post-LPS challenge. The internal organs (intestine, liver, kidney, spleen, heart and pancreas) of the piglets were obtained immediately after slaughtering and measured to calculate the relative organ weight and length to the bodyweight.

### 2.3. Blood and Tissue Sample Collections

#### 2.3.1. Blood samples.

Blood samples from the sows were collected from the auricular vein at the 16th day of lactation. Each piglet was anaesthetized with an intramuscular neck injection of pentobarbital sodium (35 mg/kg BW), and blood samples were then collected from the front cavity vein of each piglet. All blood samples were kept in heparinized tubes and were centrifuged for 10 min to separate and collect the plasma. They were stored at −80 °C for biochemical assays.

#### 2.3.2. Liver Samples of Piglets

The posterior half of liver samples were obtained immediately after slaughtering. The livers were washed in physiological saline, frozen in liquid N2, and then stored at −80 °C for real-time PCR and Western blot analysis.

### 2.4. Detection of Plasma TG, T-CHO, LDL-C, HDL-C, AKP ALT, AST, MDA, T-SOD, GSH-Px, T-AOC, and LDH

The analysis of triglyceride (TG) (Cat No. A110-1), total cholesterol (T-CHO) (Cat No. A111-1), low-density lipoprotein cholesterol (LDL-C) (Cat No. A113-1), high-density lipoprotein cholesterol (HDL-C) (Cat No. A112-1), alkaline phosphatase (AKP) (Cat No. A059-2), glutamic-pyruvic transaminase (AST), Glutamic oxaloacetylase (ALT), malondialdehyde (MDA) (Cat No. A003-1-2), total superoxide dismutase (T-SOD) (Cat No. A001-1-2), glutathione peroxidase (GSH-Px) (Cat No. A005-1-2) and total antioxidant capacity (T-AOC) (Cat No. A015-3-1) in plasma was performed according to the manufacturer’s instructions (Nanjing Jiancheng Bioengineering Institute, Nanjing, China). All absorbance values were determined with a microplate reader (Synergy 2, BioTek, Winooski, VT, USA).

### 2.5. Detection of Plasma Cytokines and Immunoglobulins

The levels of IL-1β, TNF-α, IL-6, IL-10, β-casein, IgG, IgM, sIgA, and 8-iso-prostaglandin (8-iso-PG) in plasma or milk were quantitated by commercial ELISA kits (Nanjing Jiancheng Bioengineering Institute, Nanjing, China), according to the manufacturer’s instructions, respectively.

### 2.6. Quantitative Real-Time PCR

Total RNA was extracted from liver samples by using Total RNA Kit I (50) (Cat no. R6834-01; OMEGA, Norwalk, CT, USA). One μg of RNA was employed to synthesize the first strand of cDNA in a 20 μL reverse transcription volume by using the Primescript ^TM^ RT Reagent Kit (RR047A, TaKaRa, Japan). Primers for all target genes are shown in Appendix A. Quantitative real-time PCR (RT-qPCR) was used to determine the relative expression level of target genes according to the instructions of the one-step SYBR ^®^ Premix Ex Taq (TLi RnaseH Plus) (RR420A; TaKaRa, Japan). *β*-actin was used as an internal control gene. Each sample was performed in triplicates.

### 2.7. Western Blot Analysis

The proteins in liver tissues were extracted and mixed with loading buffer as previously described by Luo et al. [14]. Equal amounts of total proteins (40 μg) were separated by 10% SDS-PAGE gels, and then transferred to a polyvinylidene difluoride (PVDF) membranes (0.45 μm pore size, IPVH00010, Millipore, Ireland Tullagreen, Carrigtwohill, County Cork, Ireland). After blocking with 5%(*w*/*v*) skimmed milk powder(Cat No. D8340, Solarbio, Shanghai, China) in TBS/0.1%Tween-20 (TBS-T) for 2 h at room temperature, membranes were washed in TBS-T three times and incubated overnight at 4 °C with primary antibodies p38 (1:2000, sc-535, SantaCruz, Dallas, TX, USA), p-p38 (1:200, sc-7973, Santa Cruz, Dallas, TX, USA), ERK1/2 (1:1000, number 9102, Cell Signaling Technology, Beverly, MA, USA), p-ERK1/2 (1:2000, number 4370, Cell Signaling Technology, Beverly, MA, USA), JNK(1:200, sc-571, Santa Cruz, Dallas, Texas, USA), p-JNK (1:500, orb10951, Biorbyt Ltd., Cambridge, UK), IκBα (1:1000, number4814, Cell Signaling Technology, Beverly, MA USA), p-IκBα (1:1000, number 9246, Cell Signaling Technology, Beverly, MA, USA) that were diluted in TBS-T with 5% (*w*/*v*) skimmed milk powder or BSA (Cat No.0218054950, ChromatoPur ^TM^, Auckland, New Zealand). After incubation overnight at 4 °C, the membranes were washed in TBS-T three times and incubated with goat anti-rabbit (1:10,000, ab97051, Abcam, Cambridge, UK) or goat anti-mouse IgG-HRP (1:2000, sc-2005, Santa Cruz, Dallas, TX, USA) antibodies for 1.5 h. The membranes were reacted with Amersham ^TM^ ECLTM Prime Western Blotting Detection Reagent (Catalogue No. RPN2232, Healthcare UK limited little Chalfont, Buckinghamshire, UK). Image acquisition was performed on the chemiluminescence detection system (Tanon, Shanghai, China). Image J software was used to quantify the density of the specific protein bands.

### 2.8. Statistical Analyses

All variables were tested for normal distribution by the Shapiro–Wilk test. Individual sow or piglet was the experimental unit for the indices. The data were analyzed by using the procedure of *t*-test in SPSS 20.0 software (SPSS Inc., Chicago, IL, USA). Because of the differences among treatments at the start of piglet study, plasma TG, IL-1β, IL-6, and IL-10 concentrations of piglet before LPS challenged were regarded as a covariate. Results were expressed as means and the standard error of the mean (SEM). A *p*-value less than 0.05 was considered to be statistically significant.

## 3. Results

### 3.1. Organ and Bodyweight of Piglets

The organ and bodyweight of piglets are shown in Table 1. There were no differences in the relative intestinal length of piglets between the SD group and the FD group. The FD did not affect the relative liver weight, pancreas weight, brain weight, spleen weight and kidney weight of piglets.

### 3.2. Effect of the Maternal Diet with Fish Oil on the Lipid Profile and Liver Function of Sows

The FD diet significantly increased the level of HDL-C in the plasma of sows compared with the SD diet (*p* < 0.05) (Table 2), but did not affect the level of TG, T-CHO and LDL-C in the plasma of sows. The level of AKP in the plasma of sows was lower in the FD group in comparison with the SD group on day 16 of lactation (*p* < 0.05). The FD diet had no significant effect on the level of AST and ALT in the plasma of sows.

### 3.3. Effect of the Maternal Diet with Fish Oil on the Oxidative Stress Parameters and Immune Cytokines in the Plasma of Sows

The FD diet significantly decreased the concentration of TNF-α, but had no significant influence on the concentrations of IL-1β, IL-6, and IL-10 in the plasma of sows on day 16 of lactation (*p* > 0.05) (Table 3). Moreover, the FD diet had no significant influence on the 8-iso-PG, T-SOD, GSH-Px, and T-AOC in sow plasma on day 16 of lactation (*p* > 0.05).

### 3.4. Effect of the Maternal Diet with Fish Oil on the Content of the Chemical Composition, Immunoglobulins, and Immune Cytokines in the Colostrum and Milk

The FD diet significantly increased the content of IgG in the colostrum (*p* < 0.05), but had no significant influence on the content of β-casein, sIgA, and IgM in the colostrum and milk (*p* > 0.05) (Table 4). The concentration of TNF-α in the milk of the FD group was lower than that of the SD group (*p* < 0.05), while the concentration of IL-10 in the milk of the FD group was higher than that in the SD group (*p* < 0.05). The FD diet did not affect the levels of IL-1β, IL-6, TNF-α, and IL-10 in the colostrum (*p* > 0.05).

### 3.5. Effect of the Maternal Diet with Fish Oil on the Lipid Profile and Liver Function in the Plasma of Piglets Pre-/Post-LPS Challenge

As shown in Table 5, the FD diet significantly decreased the level of TG in the plasma of piglets compared with the SD group on day 14 of suckling before LPS challenge (*p* < 0.05), but did not affect the levels of T-CHO, LDL-C, and HDL-C in the plasma of piglets pre-LPS challenge (*p* > 0.05). The FD diet had no significant influence on the levels of AST, ALT and AKP in the plasma of the suckling piglets pre-LPS challenge (*p* > 0.05).

The levels of T-CHO and LDL-C in the plasma of the suckling piglets from the FD group were lower than those from the SD group at 5 h post-LPS challenge (*p* < 0.05) (Table 5). Conversely, the levels of ALT and AKP in the plasma of piglets significantly increased in the FD group compared with those in the SD group at 48 h post-LPS challenge (*p* < 0.05).

### 3.6. Effect of the Maternal Diet with Fish Oil on the Oxidative Stress Status in Piglets

The activities of GSH-Px and T-AOC in the plasma of suckling piglets were higher in the FD group than that in the SD group at 48 h post-LPS challenge (*p* < 0.05) (Figure 1c,d, respectively), while the FD diet did not affect the 8-iso-PG and T-SOD in the plasma of suckling piglets pre-/post-LPS challenge (*p* > 0.05) (Figure 1a,b, respectively). Conversely, the relative expression of GSH-Px mRNA in the livers of piglets from the FD group was lower than that from the SD group at 48 h post-LPS challenge (*p* < 0.05) (Figure 1e).

### 3.7. Effect of the Maternal Diet with Fish Oil on Inflammatory Response in Piglets

The FD diet significantly increased the levels of IL-1β, IL-6, and IL-10 in the plasma of piglets before LPS challenge (*p* < 0.05) (Figure 2b–d, respectively). The level of IL-6 in the plasma of piglets was higher in the FD group than that in the SD group at 48 h post-LPS challenge (*p* < 0.05) (Figure 2c). The level of IL-1β in the plasma of piglets was significantly higher in the FD group than that in the SD group at both 5 h and 48 h post-LPS challenge (*p* < 0.05) (Figure 2b). Meanwhile, the relative expression of IL-1β mRNA in the livers of piglets was also higher in the FD group than that in the SD group at 48 h post-LPS challenge (*p* < 0.05) (Figure 2f).

### 3.8. Effect of the Maternal Diet with Fish Oil on Mechanism Parameters in Piglets

The relative expression of G-protein coupled receptor 120 (GPR120) mRNA in the livers of piglets was lower in the FD group than that in the SD group at 48 h post-LPS challenge (*p* < 0.05) (Figure 3a). Conversely, the relative expression of TGF-beta activated kinase 1 (TAK1) and NF-κB mRNA in the livers of piglets were higher in the FD group than those in the SD group at 48 h post-LPS challenge (*p* < 0.05) (Figure 3a).

The FD diet significantly down-regulated the ratio of p-ERK/ERK in the livers of piglets at 48 h post-LPS challenge (*p* < 0.05) (Figure 3b). Conversely, the ratio of p-IκBα/IκBα in the livers of piglets was significantly up-regulated in the FD group at 48 h post-LPS challenge (*p* < 0.05) (Figure 3b).

## 4. Discussion

In this study, we show that the maternal diet with fish oil might reduce the inflammatory response and improve the liver function of lactating sows, although our previous study showed that the maternal diet with fish oil did not affect reproductive performance and the growth performance of weaning piglets [12]. Our results showed that the maternal diet with fish oil increased the level of HDL-C but decreased the levels of AKP and TNF-α. These results were consistent with the reports of Papadopoulos et al. (2009) and Tanghe et al. (2013), who reported that the diet with fish oil significantly decreased the levels of TNF-α and IL-6 in the serum of sows on the 3rd and 8th day of lactation [15,16]. However, other studies found that the diet with fish oil did not affect the expression of genes related to cholesterol synthesis and absorption in the liver of lactation sows, and minor effect on the hepatic lipid metabolism [17]. Shen et al. (2015) suggested that the maternal diet with fish oil did not affect the level of cytokines in the plasma of sows [18]. The reasons for these inconsistencies may be associated with the experimental design, diet storage (vacuum dark bag storage vs. common bag storage) and the dietary antioxidant supply [19].

The maternal diet with fish oil might enhance the litter immune status by modifying both cytokines and immunoglobulins in milk [20]. The diet with fish oil resulted in the increase of antibodies in both blood and colostrum from sows after vaccination, as well as the increase in antibodies, leukocyte and IgG in the blood of piglets [21]. The maternal diet with n-3 PUFA could reduce the inflammatory stimulation of LPS on mammary gland and the levels of IL-6, IL-8, IL-1 β, and TNF-α in milk [18,22]. Our results showed that the maternal diet with fish oil increased the levels of IgG and IL-10 in the colostrum and the level of TNF-α in the milk.

Our results showed that the level of TG in the plasma of suckling piglets at the age of 14 days was reduced, but the levels of ALT, AKP, IL-1β, IL-6 and IL-10 in the plasma of suckling piglets were increased in the FD group before LPS challenge. The anti-inflammatory ability could be enhanced by IL-10, which can cross the intestinal barrier in milk and affect thymic development in progeny [23,24]. The increase of IL-10 in the plasma of suckling piglets might be transferred from milk to piglets. The increase of IL-10 in the plasma of suckling piglets indicates that the FD diet might improve the anti-inflammatory capability in the suckling piglets. Quite unexpectedly, the FD diet increased the levels of ALT and AKP in the plasma of suckling piglets at 48 h post-LPS challenge. ALT and AKP are positively correlated with liver injury [25,26,27]. In addition, the increase of IL-1β and IL-6 in the plasma of piglets pre-/post-LPS challenge indicates the increase of inflammatory response in the FD group. These results are inconsistent with the reports of Leonard et al. (2010) and McAfee et al. (2019), who reported that providing fish oil to sows’ diet from 5–7 days before delivery to the day of weaning can reduce the acute physiological stress response in the piglets at weaning day by attenuating the release of inflammatory cytokines (IL-1β, IL-6, and TNF-α) [28,29]. The reason for these inconsistencies might be related to the duration and the dose of fish oil. In the previous studies, the intake of fish oil for sows was limited before delivery, because the feeding allowance of sows was gradually decreased from 5–7 days before delivery until no feed intake on the day of farrowing in commercial pig farms. So far, the research on supplying fish oil in the diet of pig for more than four weeks is limited. Luo et al. (2013) reported that including fish oil in the diet of a sow from 10 days before delivery can reduce the expression of inflammatory cytokines in skeletal muscle and promote the growth of suckling piglets, but increase the expression of inflammatory cytokines in the spleens of weaned piglets by continuously feeding fish oil diet to weaned piglets [30].

The maternal diet with fish oil potentiated antioxidant capacity in suckled piglets. Our results showed that there was no difference in the oxidative stress state in the suckling piglets between the SD group and the FD group pre-LPS challenge. However, the FD diet increased the levels of GSH-Px and T-AOC post-48 h LPS challenge, but did not affect the oxidative stress marker products in the plasma of piglets. Furthermore, the expression of GSH-Px mRNA was decreased in piglets suckling fish oil supplemented sows. Marianne et al. (2018) reported that DHA significantly reduces oxidative stress by measures of lipid peroxidation following hypoxia-ischemia in both normothermic and hypothermic piglets [31]. Previous studies showed that placental antioxidant system has a proper capability to compensate for the oxidative stress through increasing gene expression of antioxidant enzymes in the placenta [10,32,33]. We postulate that the hepatic antioxidant system of piglets may also have an adaptive response to oxidative stress.

Interestingly, it is inconsistent that the effect of the maternal diet with fish oil on the inflammatory response and oxidative stress in piglets suckling fish oil supplemented sows. One reason might be related to the position of n-3 series double bonds, which is less susceptible to oxidative damage than the n-6 series [34]. The other reason might be related to the regulatory mechanism for the inflammatory response or oxidative stress. NF-κB plays a key role in regulating inflammation. NF-κB is sequestered in the cytoplasm in an inactive state bound to IκBα. Activation by LPS requires sequential phosphorylation of IκBα, followed by translocation of NF-κB to nucleus. The activated NF-κB is then translocated to the nucleus where it binds to specific sequences of DNA called NF-κB response elements, thus activating cytokine transcription. LPS-induced NF-κB/DNA binding was down-regulated by a low dose of EPA and DHA in THP-1 macrophages [35]. Conversely, the cytotoxicity of DHA is probably associated with inhibition of Akt and/or ERK phosphorylation in the 24 h-exposure experiment. Inhibition of ERK resulted in abrogation of IL-10 [36]. However, oxidative stress can induce ERK phosphorylation in hepatocytes, but inhibition of the phosphorylation of ERK1/2 in MAPK directly reduces ROS produced [37,38]. In our study, the maternal diet with fish oil increased the relative expression of NF-κB mRNA and the ratio of p-IκBα/IκBα proteins, but decreased the ratio of p-ERK/ERK protein in the livers of piglets at 48 h post-LPS challenge.

## 5. Conclusions

The maternal diet with fish oil might decrease the oxidative stress and inflammatory response in sows and enhance the antioxidative ability in their progenies, but might increase the susceptibility to an inflammatory response in progenies.

## Figures and Tables

**Figure 1 animals-10-01455-f001:**
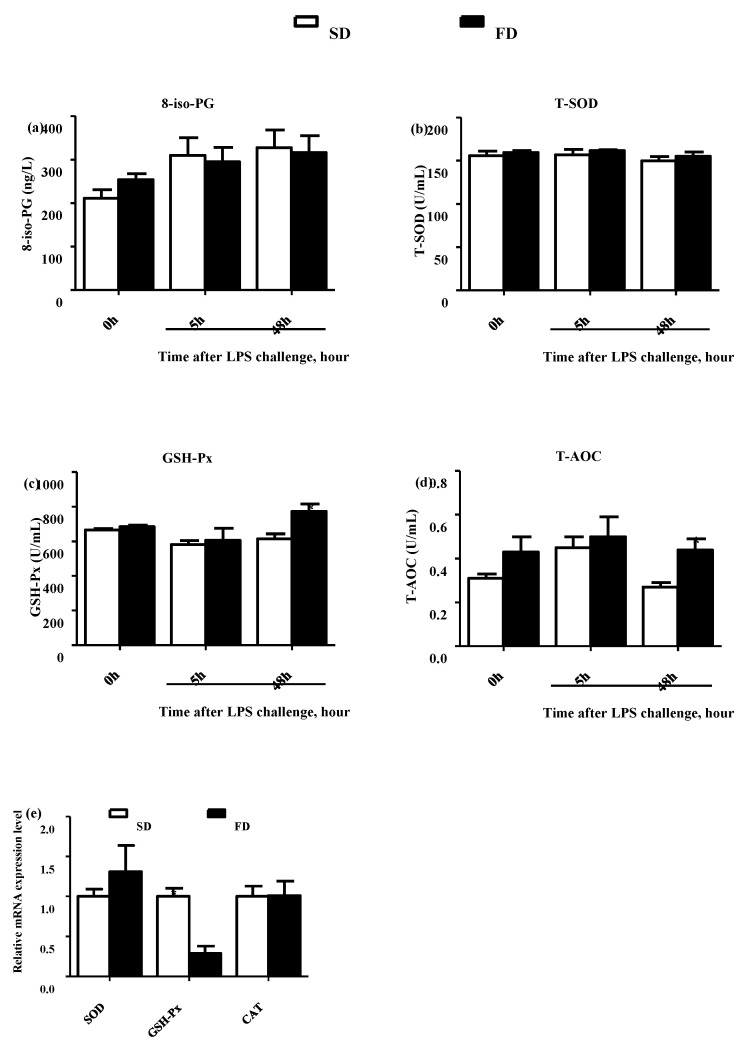
Effect of maternal diet with fish oil on the oxidative stress status in the plasma and livers of piglets. 8-iso-PG concentration (**a**), T-SOD activity (**b**), GSH-Px activity (**c**) and T-AOC (**d**) in plasma collected from piglets on 14 d (14 d), 14 + 5 h after LPS challenge (5 h), and 14 + 48 h after LPS challenge (48 h) were determined. Relative expression levels of SOD, GPx, and CAT mRNA in livers of piglets after 48 h treatment of LPS collected from sows fed different diets (**e**) were determined by real-time PCR. Values from plasma are means (*n* = 6), and values from livers are means (*n* = 4), with their standard errors represented by vertical bars. SD = soybean oil diet, FD = fish oil diet.

**Figure 2 animals-10-01455-f002:**
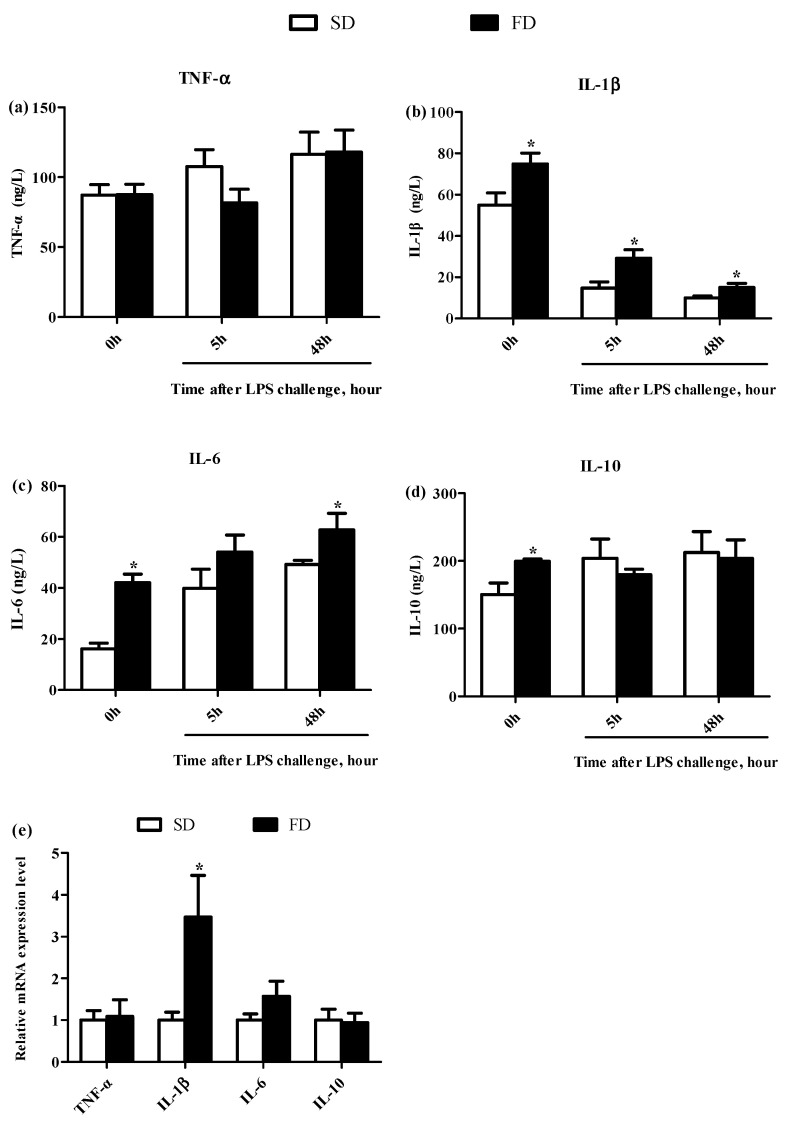
Effect of maternal diet with fish oil on the concentrations of cytokines in plasma and mRNA expression of cytokines in the livers of piglets. TNF-α concentration (**a**), IL-1β concentration (**b**), IL-6 concentration (**c**) and IL-10 concentration (**d**) in plasma collected from piglets on 14 d (14 d), 14 + 5 h after LPS challenge (5 h), and 14 + 48 h after LPS challenge (48 h) were determined. Relative expression levels of IL-1β, IL-6, TNF-α, and IL-10 mRNA in livers of piglets collected after 48 h treatment of LPS collected from sows fed different diets (**e**) were determined by real-time PCR method. Values from plasma are means (*n* = 6), and values from livers are means (*n* = 4), with their standard errors represented by vertical bars. * Mean values were significantly different between the two diet groups (*p* < 0.05). SD = soybean oil diet, FD = fish oil diet.

**Figure 3 animals-10-01455-f003:**
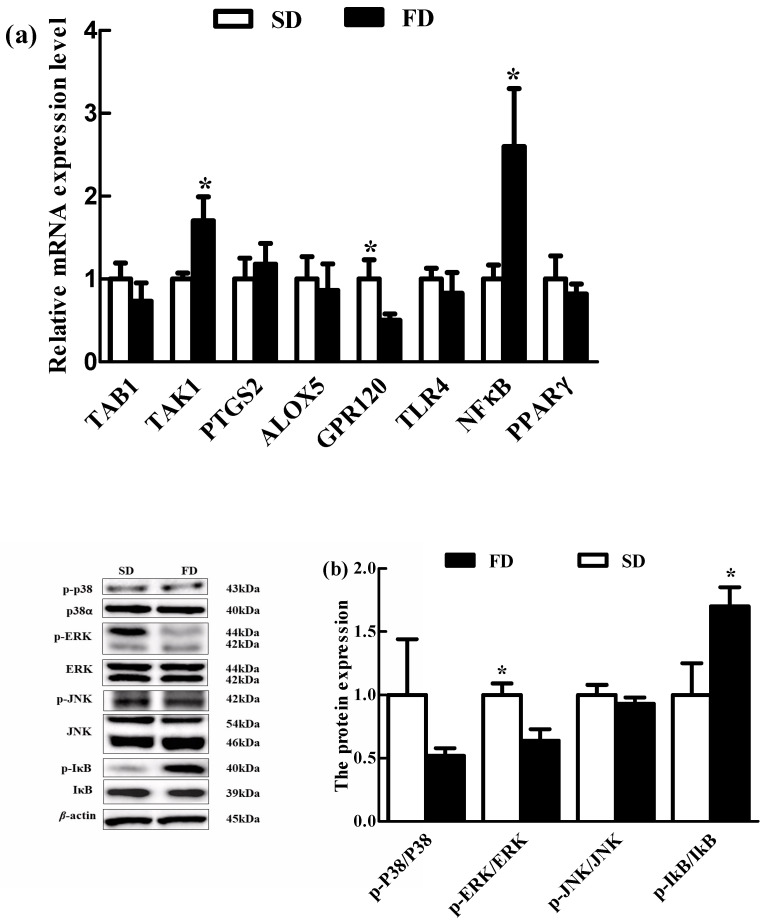
Effect of maternal diet with fish oil on mechanism parameters in the livers of piglets post-LPS challenged. (**a**) the expression of TAB1, TAK1, PTGS2, ALOX5, GPR120, TLR4, NF-κB and PPAR γ mRNA in the livers of piglets from sows fed different diets were detected by quantitative fluorescence PCR. (**b**)Western blotting method was used to detect the expression of JNK, p-JNK, p38 α, p-p38, ERK1/2, as well as phosphorylated p-ERK1/2, IκBα and p-IκBα protein in the livers of piglets from sows fed different diets. These values were expressed as the ratio of phosphorylated p-JNK, p-p38, p-ERK1/2 and P-IκB protein levels to JNK, p38, ERK1/2 and IκB protein levels. Values are means (*n* = 4), with their standard errors represented by vertical bars. * Mean values were significantly different between the two diet groups (*p* < 0.05). SD = soybean oil diet, FD = fish oil diet.

**Table 1 animals-10-01455-t001:** The bodyweight and the ratio of organ weight to bodyweight of piglets.

Item	SD	FD	*p* Value
Initial Bodyweight of Piglets (kg)	4.12 ± 0.05	4.11 ± 0.03	0.87
Slaughter Bodyweight of Piglets (kg)	4.33 ± 0.03	4.2 ± 0.08	0.20
Weight Gain (g/d)	104.2 ± 23.61	45.5 ± 44.47	0.31
Heart Weight: BW	6.51 ± 0.24	6.89 ± 0.32	0.37
Lung Weight: BW	13.82 ± 0.89	13.08 ± 0.28	0.45
Liver Weight: BW	31.88 ± 2.41	32.50 ± 1.95	0.85
Pancreas Weight: BW	1.28 ± 0.07	1.33 ± 0.06	0.60
Spleen Weight: BW	5.76 ± 0.44	6.88 ± 0.38	0.08
Kidney Weight: BW	5.76 ± 0.44	6.88 ± 0.38	0.08
Brain Weight: BW	9.46 ± 0.23	10.15 ± 0.26	0.08
Intestinal Length: BW	168.3 ± 6.12	180.9 ± 7.11	0.21

SD = soybean oil diet, FD = fish oil diet, Initial body weight of piglets = the weight of selected piglet at the age of 14 day, Slaughter body weight of piglets = the weight of selected piglet 48 h after LPS injection. Values are means ± SEM, *n* = 6.

**Table 2 animals-10-01455-t002:** Effect of the maternal diet with fish oil on the lipid profile and liver function of sows.

Item	Lactation 16 d		*p* Value
	SD	FD	
Lipid Profile			
TG(mmol/L)	0.59 ± 0.11	0.58 ± 0.13	0.92
T-CHO(mmol/L)	1.71 ± 0.26	2.14 ± 0.20	0.23
LDL-(mmol/L)	1.73 ± 0.25	1.57 ± 0.16	0.62
HDL-(mmol/L)	1.12 ± 0.10	1.56 ± 0.14 *	0.04
Liver Function			
AST(GOT) (U/L)	5.79 ± 1.45	4.30 ± 1.19	0.39
ALT(GPT) (U/L)	5.00 ± 1.05	3.36 ± 1.44	0.38
AKP (U/L)	168.4 ± 23.6 *	88.85 ± 19.82	0.85

TG = triglyceride; T-CHO = total cholesterol; LDL-C = low-density lipoprotein cholesterol; HDL-C = high-density lipoprotein cholesterol; AST = aspartate amino transferase; GPT = glutamic-pyruvic transaminase; AKP = alkaline phosphatase; NS = not significant. All results are presented as mean ± SEM (*n* = 6). * Mean values were significantly different between the two groups at the same time point (*p* < 0.05).

**Table 3 animals-10-01455-t003:** Effect of the maternal diet with fish oil on the immune cytokines and oxidative stress parameters in the plasma of sows.

Item	SD	FD	*p* Value
Cytokines			
IL-1β (ng/L)	48.45 ± 3.43	60.64 ± 8.21	0.21
IL-6 (ng/L)	67.52 ± 6.47	55.62 ± 6.74	0.25
IL-10 (ng/L)	33.33 ± 2.71	34.77 ± 3.06	0.74
TNF-α (ng/L)	322.1 ± 15.03 *	264.3 ± 11.68	0.02
Oxidative Stress Parameters			
8-iso-PG (ng/L)	336.8 ± 38.3	325.0 ± 24.0	0.80
T-SOD (U/mL)	164.2 ± 2.74	169.6 ± 3.54	0.27
GSH-Px (U/mL)	1091.4 ± 36.42	1067.2 ± 55.42	0.73
T-AOC (U/mL)	0.22 ± 0.02	0.24 ± 0.02	0.88

FD = fish oil diet; SD = soybean oil diet; IL-1β = Interleukin-1β; IL-6 = Interleukin-6; IL-10 = Interleukin-10; TNF-α = Tumor necrosis factor –alpha; 8-iso-PG = 8-iso prostaglandin; T-SOD = total superoxide dismutase; GSH-Px, glutathione peroxidase; T-AOC = total antioxidant capacity. All results are presented as mean ± SEM (*n* = 6). * Mean values were significantly different between the two groups at the same time point (*p* < 0.05).

**Table 4 animals-10-01455-t004:** Effect of the maternal diet with fish oil on the content of the chemical composition, immunoglobulins, and immune cytokines in the colostrum and milk.

	Colostrum			Milk		
	SD	FD	*p* Value	SD	FD	*p* Value
Milk profile						
β-casein (μg/mL)	26.57 ± 1.26	28.87 ± 2.31	0.38	36.22 ± 1.76	36.30 ± 2.29	0.98
IgG (mg/mL)	3.29 ± 1.45	5.47 ± 1.99 *	0.04	1.66 ± 0.35	1.79 ± 0.59	0.85
sIgA (ug/mL)	65.03 ± 2.77	52.91 ± 11.17	0.35	17.78 ± 2.20	16.42 ± 2.40	0.69
IgM (mg/mL)	15.31 ± 1.45	14.66 ± 4.42	0.89	5.53 ± 0.46	4.43 ± 0.71	0.25
Cytokines						
IL-10 (ng/L)	168.8 ± 19.9	179.31 ± 23.6	0.47	151.8 ± 6.5	185.6 ± 17.3 *	0.01
IL-1β (ng/L)	210.8 ± 16.31	187.5 ± 8.71	0.82	67.8 ± 6.00	47.8 ± 12.38	0.22
IL-6 (ng/L)	1152.1 ± 50.52	1093.8 ± 63.51	0.49	1014.1 ± 24.1	873.9 ± 66.51	0.10
TNF-α (ng/L)	234.3 ± 45.6	210.9 ± 22.9	0.77	146.3 ± 21.0 *	77.8 ± 8.33	0.02

FD = fish oil diet; SD = soybean oil diet; IgG = immunoglobulin G; sIgA = secretory immunoglobulin A; IgM = immunoglobulin M; IL-1β = interleukin-1β; IL-6 = interleukin-6; IL-10 = interleukin-10; TNF-α = tumor necrosis factor–alpha. Colostrum results are presented as mean ± SEM (*n* = 6); milk results are presented as mean ± SEM (*n* = 4). * Mean values were significantly different between the two groups at the same time point *p* < 0.05.

**Table 5 animals-10-01455-t005:** Effect of the maternal diet with fish oil on the lipid profile and liver function in the plasma of piglets pre- and post-LPS challenge.

	14 d			14 d + 5 h after LPS Challenge			14 d + 48 h after LPS Challenge		
	SD	FD	*p* Value	SD	FD	*p* Value	SD	FD	*p* Value
Lipid profile									
TG (mmol/L)	1.07 ±0.13 *	0.68 ± 0.07	0.02	0.56 ± 0.08 *	0.35 ± 0.03	0.04	0.62 ± 0.02	0.77 ± 0.08	0.1
T-CHO (mmol/L)	6.13 ± 0.85	5.19 ± 0.52	0.37	4.63 ± 0.41 *	3.61 ± 0.55	0.04	5.32 ± 0.26	4.55 ± 0.49	0.2
LDL-C (mmol/L)	3.63 ± 0.55	2.60 ± 0.31	0.14	3.51 ± 0.34*	2.46 ± 0.32	0.04	3.33 ± 0.27	2.86 ± 0.32	0.28
HDL-C (mmol/L)	3.07 ± 0.22	3.68 ± 0.31	0.14	1.72 ± 0.12	1.59 ± 0.21	0.59	2.64 ± 0.20	2.85 ± 0.17	0.45
Liver Function									
AST (U/L)	6.70 ± 0.88	8.28 ± 1.08	0.29	17.51 ± 3.54	10.45 ± 0.93	0.09	6.11 ± 0.90	6.40 ± 1.75	0.75
ALT (U/L)	8.88 ± 2.01	7.83 ± 1.48	0.69	15.03 ± 2.01	11.93 ± 0.83	0.19	9.45 ± 0.63	13.20 ± 1.41 *	0.04
AKP (U/L)	959.6 ± 87.0	1059.73 ± 74.7	0.41	1042.6 ± 33.8	1121.1 ± 54.9	0.27	646.1 ± 75.0	965. 7 ± 100.1 *	0.03

FD = fish oil diet; SD = soybean oil diet; G84d = gestation 84 days; Fd = farrowing day; L16d = lactation 16 days; TG= triglyceride; T-CHO = total cholesterol; LDL-C = low-density lipoprotein cholesterol; HDL-C = high-density lipoprotein cholesterol; AST = aspartate amino transferase; GPT= glutamic-pyruvic transaminase; AKP= alkaline phosphatase. All results are presented as mean ± SEM (*n* = 6). * Mean values were significantly different between the two groups at the same time point *p* < 0.05.

## Data Availability

The data used to support the findings of this study are included within the article and the Appendix A.

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
