# Peer review of "The Maternal Diet with Fish Oil Might Decrease the Oxidative Stress and Inflammatory Response in Sows, but Increase the Susceptibility to Inflammatory Stimulation in their Offspring"

_animals, 2020, doi:10.3390/ani10091455_

Round 1

Reviewer 1 Report

The manuscript was substantially improved. 

Results and statistical analysis are now displayed correctly.

Author Response

Thank you very much for giving us so many valuable comments.

Reviewer 2 Report

L 64-69: Please include details/procedure how the 12 sows were selected for the experiment

Author Response

Dear Professor,

  Thank you very much for giving us so many valuable comments. We had added the methods how we selected sows for our experiment in Lines 72-77 of the revised manuscript. 

  Best regards

 Wenli Luo

This manuscript is a resubmission of an earlier submission. The following is a list of the peer review reports and author responses from that submission.

Round 1

Reviewer 1 Report

The manuscript evaluated the effect of the maternal diet with fish oil on the oxidative stress and inflammatory response of sows (from gestating to lactating period). Furthermore, the work investigated the effect of the maternal diet with fish oil on the oxidative stress and inflammatory response of the progeny in the context of inflammatory stimulation (LPS).

Table 2: Is correct the value of weight gain in the FD group (piglets) 45.5 ± 44.47 ?

Remove “NS, not significant”. Not correspond.

Tables 3, 4, 5, and 6: SEM=pool SEM is not correct. Each group must have your SEM.

The manuscript needs a serious revision of the statistical analysis.

The manuscript needs to be revised by a native English speaker to improve the grammar and readability. Several sentences have mistakes.

e.g. Line 10: change “This aim of this study….” by “The aim of this study…”

Line 37: “Fish oil is rich in long chain…” add a hyphen (long-chain)

Line 51: remove “that”

Line 56-57: change the capitalization “shanghai” by “Shanghai” and “xinnong” by “Xinnong”

Line 72: change “..n-6 ploy unsaturated…” by “n-6 poly unsaturated…”

Line 79 change “…body weight is….” by “…body weight was…”

Line 81: change “Blood sample (5 mL) were….” by Blood samples (5 mL) were….”

Line 102: change “Glutamic oxaloacetylase” by Glutamic-oxalacetic transaminase

Line 172: change “Moreover, The…” by “Moreover, the….”

Line 294: ….the effects maternal diet…” by “…the effects of maternal diet…”

Line 307, 330-331: change “The reasons for these inconsistency…” by “The reasons for these inconsistencies…”

Line 310: change the verb form “enhanced” by “enhance”

Line 317, 321: change “sucking” by “suckling”

Line 335: change the verb form “were” by “was”

Line 343: change the verb form “were” by “was”

……….. among others.

Reviewer 2 Report

The objective of this paper was to test the hypothesis that maternal diet with fish oil changes the inflammatory response of sows, as well as the oxidative stress and inflammatory response of piglets before and after LPS challenge. Twelve sows were used with 6 replicates per experimental treatment. Authors concluded that maternal diet with fish oil might decrease the oxidative stress and inflammatory response, but might increase susceptibility to the inflammatory response in progenies.

The paper is interesting and would be a great contribution to swine nutrition and reproduction area in improving sow and piglet quality. Oxidative stress and inflammatory response of sows and piglets are one of the major hot topics in swine nutrition, and I believe that these topics were highlighted. However, I have major concerns in the experimental design and parameters measured that may also erroneously influence the interpretation of results.

  • The experiment lacks replicates. Yes, there were 6 replicates in the experiment. However, when conducting experiments with sows, it is expected to increase replicates because the coefficient of variation in sow traits is way greater compared with nursery and grow-finisher pig traits. Therefore, having only 6 replicates in the experiment may result in erroneous conclusions. Were the sows also blocked by weight?
  • Line 62: From which fish the fish oil was obtained? The production process of the fish oil used in the experiment must also be reported.
  • Line 66: It is indicated that diet formulations are shown in supplementary table 1. However, table 1 shows the fatty acid composition of the diets. The analyzed composition of the diets must also be reported.
  • Line 79: How was the piglet selected for blood sampling?
  • Though the focus is on oxidative stress and inflammatory response of sows and piglets, the productive and reproductive traits of sows must be added to accurately determine the effect of fish oil. Daily feed intake and backfat depth of sows must be reported. The number and body weight of pigs born alive, as well as the number of mummies, stillborn pigs, and total pigs born per litter must also be reported in sows. Pig body weight at birth or at cross-fostering and at weaning, as well as the number of days between weaning and estrus must be recorded. Average daily gain per litter and per pig, and pig survival rate from birth to weaning must also be calculated.

Reviewer 3 Report

This paper brings an interesting topic regarding the effects of fat source for sows and its litter on inflammation, added to that it could contribute as a model for human research.

The English must be reviewed.

Line 10. The Abstract must be improved, a lot of important information are missing, as an example on line 14 you say blood sample of the sows were collected, but do not describe at which moment.

Line 30. Do not use “Sow” and “Oxidative stress” as keywords, they are already on the title.

Introduction must be improved.

Line 42. Cytokines on the milk are not good? This is not correct…

Line 51 to 57. Font size is bigger.

Line 55. Number of the protocol must be provided.

 Line 66. Diet formulations are shown in Supplementary Table 1. à This is not true, Table 1 on the supplementary material is Fatty acid composition of the oils, the diet formulation table must be provided.

Line 90. When blood of the sows was collected?

Lines 93-97. The samples were snaped frozen on nitrogenous?

Lines 98-106. Some of the variables described here are not to determine oxidant and antioxidant capacity.

Lines 146. What do you mean with organ indices, I do not believe this is the best name.

Table 2 – You must explain better what is initial weight, weigh gain in how many days? Hours?

Table 3, 4, and 6  – You must present P values for all variables.

Line 294-296. Erase

The discussion section must be better eplored.

Line 379. Ethics approval - Number of the protocol must be provided.